# Image Fundus Classification System for Diabetic Retinopathy Stage Detection Using Hybrid CNN-DELM

**Dian Candra Rini Novitasari** [1,2,*], **Fatmawati Fatmawati** [2,*], **Rimuljo Hendradi** [2], **Hetty Rohayani** [3], **Rinda Nariswari** [4], **Arnita Arnita** [5], **Moch Irfan Hadi** [6], **Rizal Amegia Saputra** [7] and **Ardhin Primadewi** [8]

[1]  Mathematics Department, Faculty of Science and Technology, UIN Sunan Ampel Surabaya, Surabaya 36743, Indonesia
[2]  Mathematics Department, Faculty of Science and Technology, Universitas Airlangga, Surabaya 36743, Indonesia
[3]  Sains and Technology Faculty, Muhammadiyah Jambi University, Jambi 23265, Indonesia
[4]  Statistics Department, School of Computer Science, Bina Nusantara University, Jakarta 10110, Indonesia
[5]  Department of Mathematics, Universitas Negeri Medan, Medan 20028, Indonesia
[6]  Biology Department, Faculty of Science and Technology, UIN Sunan Ampel Surabaya, Surabaya 36743, Indonesia
[7]  Information System Department, Faculty of Engineering and Informatics, Universitas Bina Sarana Informatika, Sukabumi 43111, Indonesia
[8]  Department of Informatics Engineering, Muhammadiyah University of Magelang, Magelang 56111, Indonesia
*  Correspondence: diancrini@uinsby.ac.id (D.C.R.N.); fatmawati@fst.unair.ac.id (F.F.)

**Abstract:** Diabetic retinopathy is the leading cause of blindness suffered by working-age adults. The increase in the population diagnosed with DR can be prevented by screening and early treatment of eye damage. This screening process can be conducted by utilizing deep learning techniques. In this study, the detection of DR severity was carried out using the hybrid CNN-DELM method (CDELM). The CNN architectures used were ResNet-18, ResNet-50, ResNet-101, GoogleNet, and DenseNet. The learning outcome features were further classified using the DELM algorithm. The comparison of CNN architecture aimed to find the best CNN architecture for fundus image features extraction. This research also compared the effect of using the kernel function on the performance of DELM in fundus image classification. All experiments using CDELM showed maximum results, with an accuracy of 100% in the DRIVE data and the two-class MESSIDOR data. Meanwhile, the best results obtained in the MESSIDOR 4 class data reached 98.20%. The advantage of the DELM method compared to the conventional CNN method is that the training time duration is much shorter. CNN takes an average of 30 min for training, while the CDELM method takes only an average of 2.5 min. Based on the value of accuracy and duration of training time, the CDELM method had better performance than the conventional CNN method.

**Keywords:** diabetic retinopathy; CNN architecture; feature learning; DELM classification

## 1. Introduction

Diabetic retinopathy (DR) is an eye disease caused by high blood sugar levels that attack the retinal capillaries of the eye [1]. DR is the leading cause of blindness suffered by working-age adults [2,3]. In 2015 blindness due to DR was estimated to reach 2.6 million people and the projected results on the number of people with DR in 2020 reached 3.2 million [1]. The prevalence of DR is estimated to increase in the next decade along with the increase in diabetes, especially in Asian countries, such as Indonesia, India, and China [4]. The increasing prevalence of DR can be prevented by early detection and treatment of eye damage caused by DR [5]. A rapid screening process is required so that people with DR receive immediate and appropriate treatment [6]. This screening process can be carried out by utilizing technological advances, that is, the Computer Aided Diagnosis (CAD) system. CAD system development can diagnose DR efficiently. High

computational mechanisms enable better diagnostic capabilities by classifying fundus image data to identify damage to the retina [7]. CAD begins with pre-processing image data to optimize data in the learning model [8].

CAD in diagnosing DR has been widely used in previous studies. DR detection by applying the Gray Level Co-Occurrence Matrix (GLCM) and Support Vector Machine (SVM) methods were conducted to obtain accurate results. Accurate results in DR and normal diagnoses reached 82.5%, while in PDR and NPDR diagnoses, the accuracy reached 100% [9]. One of the difficulties in DR detection is studying the features in the fundus image [7]. The process of studying features can be carried out by implementing the convolutional neural network (CNN) algorithm [10]. A study by Navoneel Chakrabarty showed that the CNN algorithm produced high accuracy in the DR classification process, which was 100% [11].

CNN is a deep learning algorithm that has several different architectures. The architectures in the CNN algorithm include GoogleNet, ResNet, and DenseNet [12]. Previous researchers have widely used the architectures in the CNN algorithm. A research by R Anand used the GoogleNet architecture on the CNN algorithm to detect faces. The overall accuracy result reached 91.43%, which was quite high for facial recognition and more than conventional Machine Learning (ML) techniques. The number of data trained on the model also affects the accuracy of a classification system. The more data trained with optimal computing power, the more accurate the prediction results, up to 99% [13]. Arpana Mahajan has studied ResNet architecture to observe the features of categorical images. The features that have been studied are further classified using SVM. ResNet architecture was tested based on the number of layers. The test results showed that ResNet with 18 layers obtained a higher accuracy of 93.57% [14]. Research conducted by Hua Li used DenseNet architecture to classify benign and malignant mammogram images. The accuracy obtained from the DenseNet architecture reached 94.55% [15]. Based on previous studies, the CNN algorithm performed well in the classification process and had many layers. A large number of layers in the CNN algorithm required a computer with a large capacity and a long duration of the training process [16]. Several other researchers have tried to develop CNN to overcome these problems by changing the existing classification system on CNN to form a developed method that uses the convolutional features in the CNN architecture but uses a different classification method.

The development of CNN methods, such as the Convolutional Extreme Learning Machine (CELM), which uses the convolutional features in the CNN architecture and the Extreme Learning Machine classification method. Research on CELM was conducted to identify handwritten MNIST dataset [17–19]. The results of this study indicated that the accuracy obtained by CELM was better than ELM and CNN, that is, 98.43%, with a training time faster than CNN and ELM. Although CELM is better than CNN, basically, ELM is still a single hidden layer method and is still not good at pattern recognition in big data; so, the development of the ELM method by applying a multilayer and deep learning system is called Deep Extreme Learning Machine (DELM) [20]. The DELM method has several advantages, especially in training time, making it one of the deep learning methods with the fastest training process. DELM also has good results in terms of image classification (MNIST database, CIFAR-10 dataset, and Google Streetview House Number dataset) with an average accuracy of 95.16% [21]. The DELM algorithm can produce a high accuracy in just 9.02 s [22]. DELM is a combination of several algorithms that are the result of the development of the Extreme Learning Machine (ELM) algorithm. DELM has a more complex structure than ELM, but DELM can train models faster than the ELM algorithm [23].

Based on the description of the problems, the convolutional features in the CNN architecture can well recognize the pattern of an image. Therefore, in the classification process by DELM, it is necessary to use the convolutional features in the CNN architecture for feature extraction. This study aims to build a Hybrid CNN-DELM or Convolutional Deep Extreme Learning Machine (CDELM) method that can collectively recognize image

patterns to produce performance for better accuracy and faster training time. The CDELM method is applied in this study using the CNN architectures, that is GoogleNet, ResNet, and DenseNet.

## 2. Materials and Methods

This study implements the CNN and DELM methods to detect eye damage due to DR based on fundus images. This study used fundus image data obtained from the DRIVE and MESSIDOR databases. Each database consisted of 44 and 1200 fundus images of 4 classes (normal, mild, moderate, and severe). Based on microvascular lesion type, number, and location, the severity level of DR is classified into three stages: Mild, Moderate, and Severe [24]. The mild stage has a microaneurysm with or without retinal hemorrhage. In the moderate stage, there is a microaneurysm with retinal hemorrhages and white, cotton-like spots. The severe stage occurs with profuse bleeding in four quadrants of the retina and white spots in two or more quadrants [25]. The difference in fundus images at each stage can be seen in Figure 1.

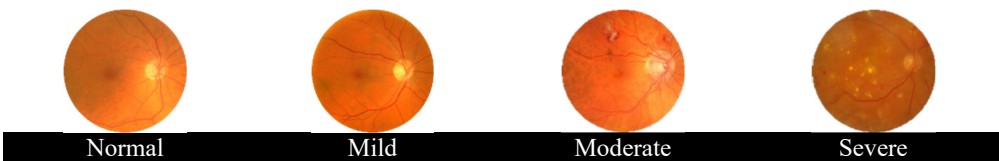

**Figure 1.** Stages of DR.

Each database was augmented with random rotation at 1 degree to 359 degrees; it amounted to 13,032, with 3258 fundus images in each class. The rotation method is suitable if applied to images with features, such as circles. The fundus image has circular features, so applying the rotation method for the augmentation process is more efficient. The next stage is image enhancement using the CLAHE method to clarify features in the fundus image. The features in the fundus image are extracted by applying the CNN method's feature learning concept. The graphical abstract in this study can be seen in Figure 2.

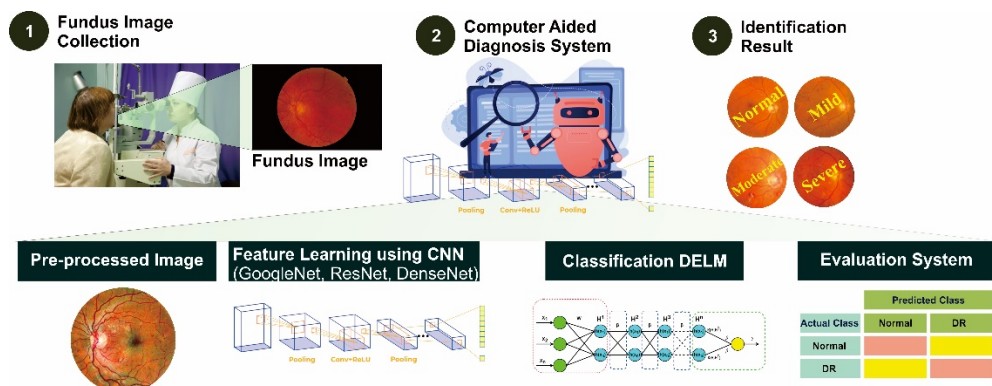

**Figure 2.** Graphical abstract of DR classification using CDELM method.

Based on Figure 2, data preprocessing is to change the image data size to 224 × 224 and to increase the variance of the image data by augmentation process. The process of transforming image data into numerical data as image features is carried out by utilizing the CNN algorithm in which the features of the image data will be extracted. The CNN algorithm has several architectural structures that can be used for feature learning, such as GoogleNet, ResNet, and DenseNet. The performance of the three architectures is compared to find out which architecture is the best in studying features in the fundus image. The features generated from the feature learning process are then classified using the DELM algorithm. The classification process consists of training and testing processes. In the training process, it produces a model that will be validated in the testing process and evaluated using a confusion matrix.

## 2.1. Diabetic Retinopathy

Diabetic retinopathy (DR) is a complication that often occurs in diabetes mellitus (DM) patients. DR causes damage to the retina of the eye involving pathological blood vessels. Increased blood sugar can increase the risk of damage to blood vessels in the retina and interfere with the patient's visual system [12,26]. DR can cause visual impairment to blindness in patients [27]. DR can be identified using the patient's fundus image. It is characterized by bleeding, hard exudate, and cotton spots [28]. The non-proliferative type of DR is characterized by microaneurysms, hard exudates, hemorrhages, and venous abnormalities [29].

## 2.2. Convolutional Neural Network

Convolutional neural network (CNN) is an effective machine learning technique for deep learning [30]. CNN is widely applied in medical image analysis research because CNN maintains spatial relationships when filtering input images [31]. In recognizing image features, CNN applies a feed-forward neural network with each unit in the adjacent layer fully connected. CNN has a particular layer where only specific units are connected to fully connected layers that will classify the image features [32]. In an overview of the CNN algorithm architecture, as shown in Figure 3, it can be seen that CNN consists of feature learning and classification [33,34].

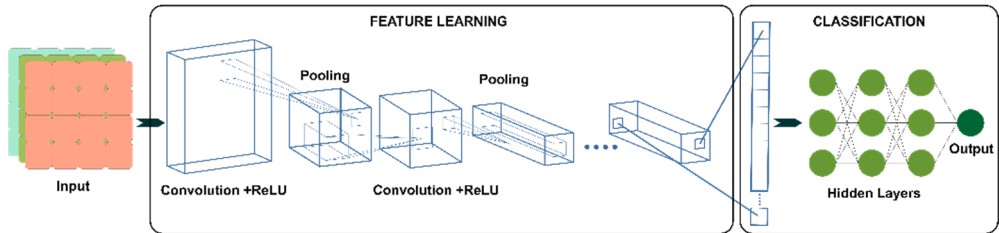

**Figure 3.** CNN architecture.

In this research, the feature learning process aims to extract the feature of images. CNN uses the convolution layer as a primary layer for image feature extraction. Feature extraction by convolution operation with a filter across every local area in the image to learn a spatial pattern of the image. The output of the convolutional process is called feature maps which will be processed in the next layer [33]. After the convolution process, features maps are simplified using pooling layers [35]. The feature maps pass the ReLU layer as an activation function to reduce the time in the convergence of stochastic gradient descent [36]. The feature learning process in the CNN algorithm can be carried out with several different architectures. The architectures in the CNN algorithm include GoogleNet, ResNet, and DenseNet [12].

### 2.2.1. GoogleNet

GoogleNet is a transfer learning that implements inception, which can improve computing performance. In the ILSVRC competition, GoogleNet earned 5.5% with a top five rating in classification performance [37]. GoogleNet, developed by Google, trains a network with millions of images to identify thousands of daily life images. GoogleNet has 144 layers, and the input image data on GoogleNet is 224 × 224 × 3 [38]. The difference between GoogleNet and conventional CNN is in the inception module. GoogleNet implements the inception module that utilizes small convolutions to reduce the number of parameters [39]. An illustration of GoogleNet's architecture is shown in Figure 4.

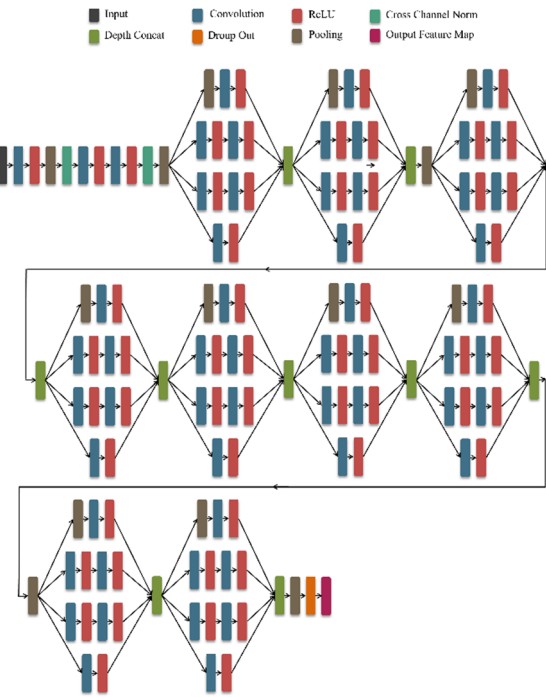

**Figure 4.** GoogleNet architecture.

### 2.2.2. ResNet

ResNet is a CNN residual developed by the Microsoft team. ResNet won first place in the largest ImageNet visual recognition competition ILSVRC [40]. The ResNet model architecture is based on residual training. ResNet uses an image input measuring $224 \times 224$ with three layers, namely Red (R), Green (G), and Blue (B) layers [37]. The ResNet model does not learn all the features but only some of the features that are identified as residuals. Residuals are reduced features known from the input layer [41]. Illustration of ResNet architecture shows in Figure 5.

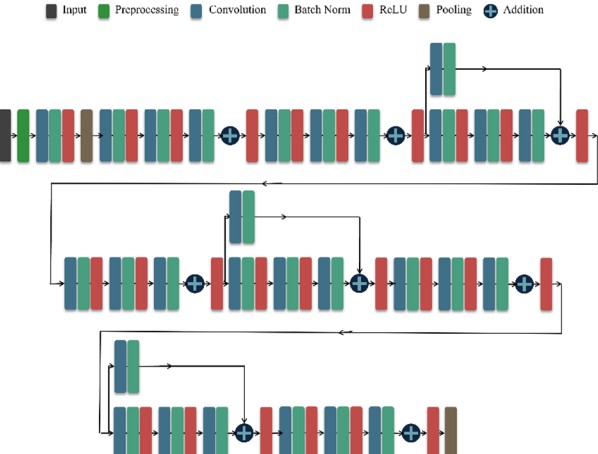

**Figure 5.** ResNet architecture.

### 2.2.3. DenseNet

DenseNet is a CNN architecture that has dense connections. DenseNet is a development of ResNet, which has a network structure which is more innovative, simple, and effective [42]. Connections between layers in DenseNet architecture reduce the parameters needed to study feature maps by eliminating those that are unnecessary. The advantages of the DenseNet architecture are that it can handle missing gradients, strengthen feature

propagation, reuse features used, and substantially reduce parameters [41]. An illustration of DenseNet architecture is shown in Figure 6.

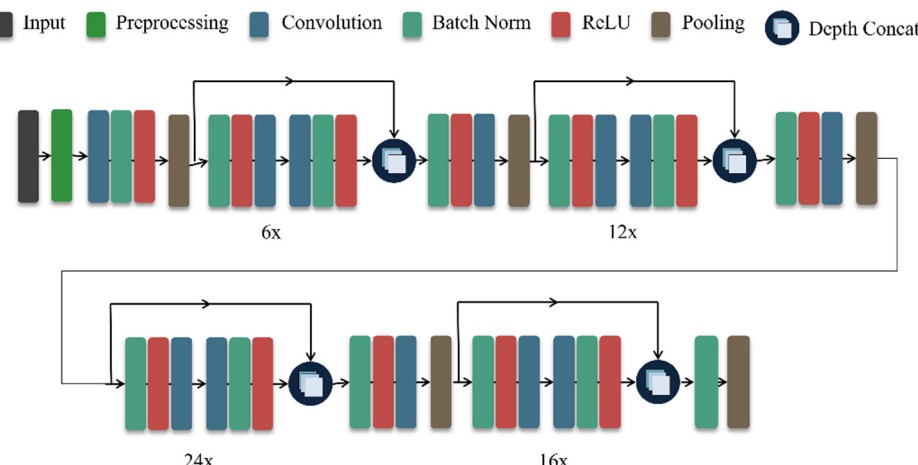

**Figure 6.** DenseNet architecture.

### 2.3. Deep Extreme Learning Machine (DELM)

Extreme Learning Machine (ELM) is one of the algorithms introduced by Guang-Bin Huang [43] as a continuation of the Single Layer Forward Network (SLFN). The hidden network in the ELM algorithm is generated randomly, and the weights are calculated based on the least squares solution. The ELM algorithm, with the addition of a layer on the network, was developed into the Deep Extreme Learning Machine (DELM) algorithm [44]. The training method on the DELM algorithm is better in prediction than conventional neural networks. The DELM model architecture has three layers: the input layer, hidden layer, and output layer [45].

DELM combines several modified ELM methods, namely Multilayer Extreme Learning Machine (MELM) and Kernel Extreme Learning Machine (KELM), where the output of the MELM algorithm will be the input to the KELM algorithm. The MELM algorithm comprises an Extreme Learning Machine Auto Encoder (ELM-AE) stack. The training of parameters in each layer in the DELM algorithm is carried out by implementing the MELM algorithm. The kernel function in the KELM algorithm is used to determine whether the output of MELM is linear or non-linear. The output of the MELM algorithm is $\mathbf{H}_n$, which is then used to build a kernel function with $\mathbf{H}_{n+1}\mathbf{H}_{n+1}{}^T$ [23]. The DELM algorithm architecture can be seen in Figure 7.

Based on the DELM architecture illustrated in Figure 5, the input layer processes the input data using the basic ELM algorithm and produces an output value in the first hidden layer. Calculations on the 2nd hidden layer to the $(n-1)$ hidden layer are carried out using the ELM-AE algorithm arrangement that forms MELM. The output of the MELM algorithm is streamed to the nth hidden layer, where there is a kernel function to produce the final result of the DELM algorithm. The output of the first hidden layer is calculated using Equation (1).

$$\mathbf{H} = g\left(\mathbf{W}^{\mathrm{T}} X + \mathbf{b}\right) \tag{1}$$

where $\mathbf{W}$ is the weight linking the input layer and the hidden layer, $\mathbf{b}$ is the bias of the hidden layer, and $g(x)$ is the activation function. While the calculation of the nth hidden layer is calculated using Equation (2).

$$\mathbf{H}^{\mathrm{n}} = g(\mathbf{H}_{n-1}(\boldsymbol{\beta}_{n-1})^{\mathrm{T}}) \tag{2}$$

where $\beta$ is the output weight value. The $\beta$ value is determined based on the number of nodes and input nodes, as formulated in Equations (3)–(5).

$$\beta^* = \left(\mathbf{H}^T\mathbf{H} + \frac{\mathbf{I}_{n_h}}{c}\right)^{-1}\mathbf{H}^T\mathbf{X}, \; n_i > n_h \tag{3}$$

$$\beta^* = \mathbf{H}^T\left(\mathbf{H}\mathbf{H}^T + \frac{\mathbf{I}_{n_i}}{c}\right)^{-1}\mathbf{X}, \; n_i < n_h \tag{4}$$

$$\beta^* = \mathbf{H}^T\left(\mathbf{H}\mathbf{H}^T\right)^{-1}\mathbf{X}, \; n_i = n_h \tag{5}$$

where c is the scale parameter, $\mathbf{I}_{n_h}$ is the identity matrix with dimension $n_h$, $\mathbf{I}_N$ is the identity matrix with dimension $\mathbf{N}$, $n_h$ and $n_i$ are the nodes and input nodes in the hidden layer. The calculation is carried out until the hidden layer is calculated (n − 1). The n-th hidden layer is calculated based on the value in the (n − 1) hidden layer as the new input value defined in Equation (6).

$$\mathbf{X}_{new} = \mathbf{X}\beta^T \tag{6}$$

Calculating the nth hidden layer to get the output value (**Y**) from the DELM algorithm is carried out using Equation (7).

$$Y = \left[\left(K(\mathbf{H}, \mathbf{H}^T)\right)^T\right]^{-1} \cdot \left(\frac{\mathbf{I}}{c} + \mathbf{H}\mathbf{H}^T\beta\right)s \tag{7}$$

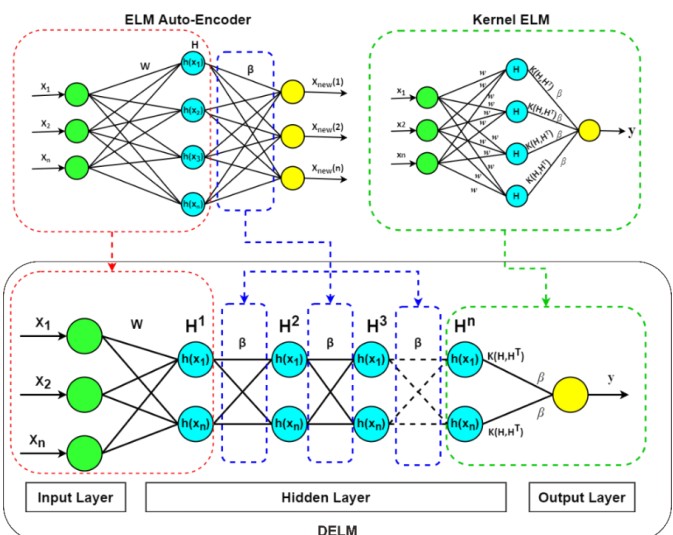

**Figure 7.** DELM Architecture.

## 3. Results

In this research, classifications were made with two multi-class experiments: 2-class (Normal and DR) and 4-class (Normal, Mild, Moderate, and Severe). The initial stage of the research is cropping the image, CLAHE, resizing according to the CNN architecture input size, and performing the augmentation process. The results of the CLAHE process are shown in Figure 8.

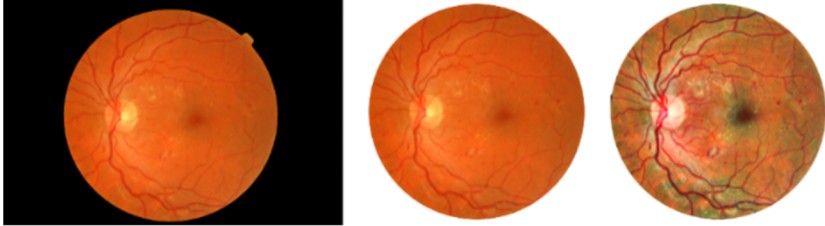

**Figure 8.** Original Fundus Image, Result of Cropping Image, Result of CLAHE.

Based on the results of the CLAHE process in Figure 6, it shows that there is an increase in image quality. DR disease is highly dependent on the state of the blood vessels in the retina. The CLAHE method can clearly show the condition of the blood vessels and identify the presence of microaneurysm to bleeding in the retina. The following preprocessing stage is resizing and augmentation. The augmentation method used was a random rotation from 1 degree to 359 degrees. Augmentation results can be seen in Figure 9.

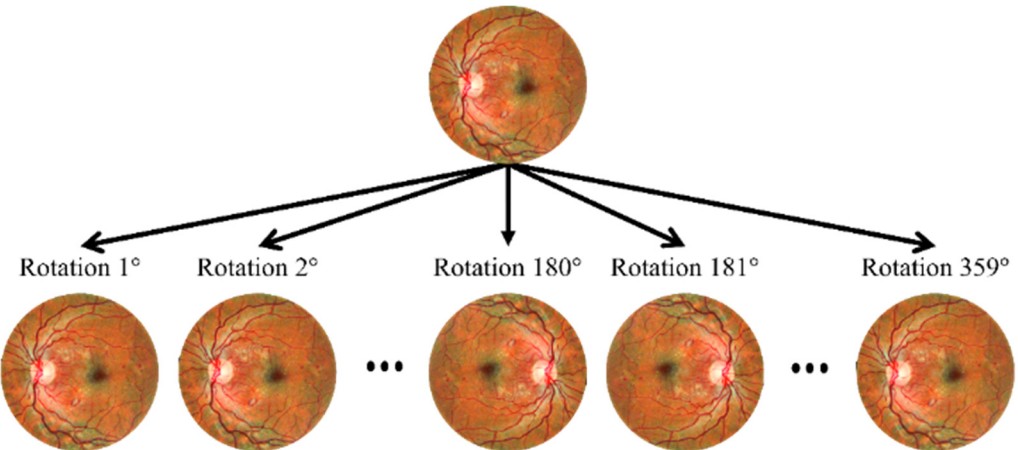

**Figure 9.** Image Augmentation with Rotation 1°.

The next stage after augmentation is feature extraction from the fundus image. This stage is carried out by comparing several CNN architectures: GoogleNet, ResNet, and DenseNet. The feature extraction results on each CNN architecture can be seen in Table 1. The results of CNN feature extraction are obtained from feature learning through several convolution processes, pooling, and applying activation functions. These steps are repeated until a vector of the extraction results from each image data is obtained.

Based on Table 1, the results of feature extraction from each architecture show different values and features. The data is divided into training and testing data using five-fold cross-validation. In the two-class dataset, training data contains 2880 images in each class and 720 images in each class for testing data. While in the four-class dataset, training data has 2607 images in each class and 651 images in each class for the testing dataset. Next is the classification process using the DELM method, which is evaluated based on accuracy, sensitivity, specificity, and duration of training time. It is used to determine the most optimal model of the DR classification system. Then, CDELM performance on the 2-class and 4-class DRIVE and MESSIDOR data is compared. The results of experiment conducted on the 2-class DRIVE and MESSIDOR data on several CNN architectures are shown in Tables 2 and 3. The bold values in Tables 2–5 mean the best result of the kernel experiment.

**Table 1.** Results on Feature Extraction CNN Architecture from A Fundus Image.

| No Feature | GoogleNet | ResNet18 | ResNet50 | ResNet101 | DenseNet |
|:---:|:---:|:---:|:---:|:---:|:---:|
| 1 | 0.0500 | 1.2727 | 3.1410 | 2.0698 | −0.0001 |
| 2 | 0.2205 | 1.7893 | 0.0496 | 0 | 0.0004 |
| 3 | 0 | 0.3892 | 0 | 0.0629 | 0.0018 |
| 4 | 0.5618 | 0.2355 | 0.1208 | 0 | −0.0963 |
| 5 | 0.0173 | 0.1450 | 0.0105 | 0.1196 | 0.0022 |
| 6 | 0.0241 | 0.6516 | 0 | 0.4925 | 0.0002 |
| 7 | 0 | 0.3554 | 0.0048 | 1.5753 | −0.0004 |
| 8 | 0 | 1.0218 | 0.2059 | 0.1258 | 0.0007 |
| 9 | 0.0268 | 1.3834 | 0 | 0.3570 | 0.0002 |
| 10 | 0.1011 | 0.7051 | 0.1300 | 0.5861 | 0.0090 |
| ⋮ | ⋮ | ⋮ | ⋮ | ⋮ | ⋮ |
| feature-n | 0.4129 | 0.0715 | 0.3104 | 0.4698 | 0.8898 |
| Total feature | 1024 | 512 | 2048 | 2048 | 1920 |

**Table 2.** Results of Experiment on Two-Class DRIVE Dataset.

| CNN Architecture | Kernel | Accuracy (%) | Sensitivity (%) | Specificity (%) | Duration (s) |
|:---:|:---:|:---:|:---:|:---:|:---:|
| ResNet18 | Linear | 90.83 | 90.83 | 90.97 | 288.50 |
| | RBF | 92.78 | 92.78 | 92.92 | 288.10 |
| | **Poly** | **100.00** | **100.00** | **100.00** | **291.22** |
| ResNet50 | Linear | 95.90 | 95.90 | 96.21 | 299.17 |
| | RBF | 98.40 | 98.40 | 98.45 | 295.58 |
| | **Poly** | **100.00** | **100.00** | **100.00** | **295.37** |
| ResNet101 | Linear | 98.06 | 98.06 | 98.12 | 296.63 |
| | RBF | 99.44 | 99.44 | 99.45 | 300.62 |
| | **Poly** | **100.00** | **100.00** | **100.00** | **298.02** |
| GoogleNet | Linear | 91.11 | 91.11 | 91.92 | 300.24 |
| | RBF | 97.36 | 97.36 | 97.45 | 299.29 |
| | **Poly** | **100.00** | **100.00** | **100.00** | **307.13** |
| DenseNet | Linear | 96.25 | 96.25 | 96.35 | 302.45 |
| | RBF | 97.36 | 97.36 | 97.49 | 328.36 |
| | **Poly** | **100.00** | **100.00** | **100.00** | **306.15** |

**Table 3.** Results of Experiment on Two-Class MESSIDOR Dataset.

| CNN Architecture | Kernel | Accuracy (%) | Sensitivity (%) | Specificity (%) | Duration (s) |
|---|---|---|---|---|---|
| ResNet18 | Linear | 90.56 | 90.56 | 90.72 | 290.25 |
| | RBF | 92.78 | 92.78 | 92.92 | 284.36 |
| | **Poly** | **100.00** | **100.00** | **100.00** | **287.24** |
| ResNet50 | Linear | 96.60 | 96.60 | 96.78 | 293.76 |
| | RBF | 98.33 | 98.33 | 98.38 | 291.99 |
| | **Poly** | **100.00** | **100.00** | **100.00** | **294.61** |
| ResNet101 | Linear | 97.64 | 97.64 | 97.72 | 298.70 |
| | RBF | 99.44 | 99.44 | 99.45 | 293.67 |
| | **Poly** | **100.00** | **100.00** | **100.00** | **293.08** |
| GoogleNet | Linear | 92.22 | 92.22 | 92.86 | 294.98 |
| | RBF | 96.94 | 96.94 | 97.06 | 288.39 |
| | **Poly** | **100.00** | **100.00** | **100.00** | **290.46** |
| DenseNet | Linear | 95.49 | 95.49 | 95.62 | 298.19 |
| | RBF | 97.29 | 97.29 | 97.43 | 290.27 |
| | **Poly** | **100.00** | **100.00** | **100.00** | **289.96** |

**Table 4.** Results of Experiment on Four-Class DRIVE Dataset.

| CNN Architecture | Kernel | Accuracy (%) | Sensitivity (%) | Specificity (%) | Duration (s) |
|---|---|---|---|---|---|
| ResNet18 | Linear | 79.72 | 79.72 | 79.85 | 136.04 |
| | RBF | 84.33 | 84.33 | 84.33 | 135.43 |
| | **Poly** | **100.00** | **100.00** | **100.00** | **140.27** |
| ResNet50 | Linear | 88.52 | 88.52 | 88.83 | 136.95 |
| | RBF | 95.51 | 95.51 | 95.63 | 138.10 |
| | **Poly** | **100.00** | **100.00** | **100.00** | **146.84** |
| ResNet101 | Linear | 93.55 | 93.55 | 93.63 | 137.25 |
| | RBF | 98.00 | 98.00 | 98.00 | 138.84 |
| | **Poly** | **100.00** | **100.00** | **100.00** | **142.72** |
| GoogleNet | Linear | 82.11 | 82.11 | 82.29 | 136.07 |
| | RBF | 95.12 | 95.12 | 95.18 | 138.04 |
| | **Poly** | **100.00** | **100.00** | **100.00** | **141.78** |
| DenseNet | Linear | 90.52 | 90.52 | 90.60 | 138.41 |
| | RBF | 95.47 | 95.47 | 95.46 | 138.37 |
| | **Poly** | **100.00** | **100.00** | **100.00** | **140.99** |

**Table 5.** Results of Experiment on 4 Class MESSIDOR Dataset.

| CNN Architecture | Kernel | Accuracy (%) | Sensitivity (%) | Specificity (%) | Duration (s) |
|---|---|---|---|---|---|
| ResNet18 | Linear | 68.28 | 68.28 | 69.11 | 137.21 |
| | RBF | 68.20 | 68.20 | 67.72 | 138.00 |
| | **Poly** | **93.70** | **93.70** | **93.70** | **143.55** |
| ResNet50 | Linear | 61.94 | 61.94 | 61.05 | 139.06 |
| | RBF | 67.13 | 67.13 | 66.58 | 144.96 |
| | **Poly** | **97.77** | **97.77** | **97.77** | **144.78** |
| ResNet101 | Linear | 62.44 | 62.44 | 62.44 | 159.63 |
| | RBF | 67.32 | 67.32 | 67.10 | 161.81 |
| | **Poly** | **98.20** | **98.20** | **98.19** | **166.94** |
| GoogleNet | Linear | 60.14 | 60.14 | 59.36 | 161.97 |
| | RBF | 64.44 | 64.44 | 64.19 | 140.73 |
| | **Poly** | **95.97** | **95.97** | **96.03** | **144.16** |
| DenseNet | Linear | 63.56 | 63.56 | 63.17 | 137.56 |
| | RBF | 64.82 | 64.82 | 64.39 | 162.28 |
| | **Poly** | **97.89** | **97.89** | **97.90** | **142.77** |

The features obtained from several CNN architectures, such as ResNet-18, ResNet-50, ResNet-101, GoogleNet, and DenseNet, can represent each class in the 2-class DRIVE data by the accuracy value, which reaches more than 90%. The high accuracy value is also influenced by the performance of DELM as a classification method that can classify many features well. Another advantage of the DELM classification method is that it can be seen from the duration of the training time, which is less than 5 min. The performance of the DELM method is highly dependent on the compatibility of the data with the kernel functions used. In the 2-class DRIVE data, the data shows that the polynomial kernel function performs better than the linear kernel or the RBF kernel. The graph to compare the accuracy values of the two-class DRIVE dataset is shown in Figure 10.

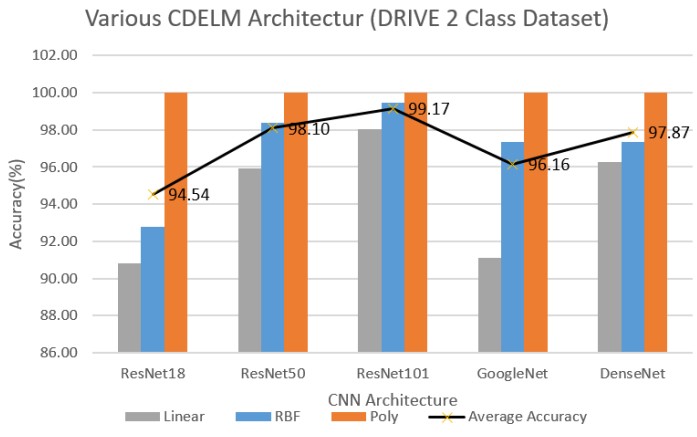

**Figure 10.** Results of Experiment on 2-class DRIVE Dataset.

Based on Figure 10, using a polynomial kernel in the DELM classification can achieve 100% accuracy in each architecture. The DELM classification system using a linear kernel only obtains accuracy in the range of 90% to 98%. Compared to the linear kernel, the RBF kernel is able to better separate data from each class, with an accuracy value of 92% to 99%. As the best CNN architecture in terms of the average accuracy of each architecture, ResNet-

101 architecture can represent the characteristics of the normal fundus and DR images with the best accuracy value of 99.17%, followed by ResN=et-50, DenseNet, GoogleNet, and ResNet-18 architectures, with an average accuracy of 98.10%, 97.87%, 96.16%, and 94.54%, respectively.

Similar to the previous experiment, in the two-class MESSIDOR dataset, the CDELM methods also perform well, with an accuracy above 90%. Therefore, it shows that the CNN architecture can well represent the fundus image features in the two-class MESSIDOR dataset. The combination of a suitable CNN method in extracting features with the DELM method, which has good performance in classification, plays an essential role in producing high accuracy in the classification system. The DELM classification process takes only approximately 4 min. The duration of the training is relatively shorter than the 2-class DRIVE 2 dataset experiment, which is 1 min faster than the previous. The graph on the accuracy value comparison from the two-class MESSIDOR dataset is shown in Figure 11.

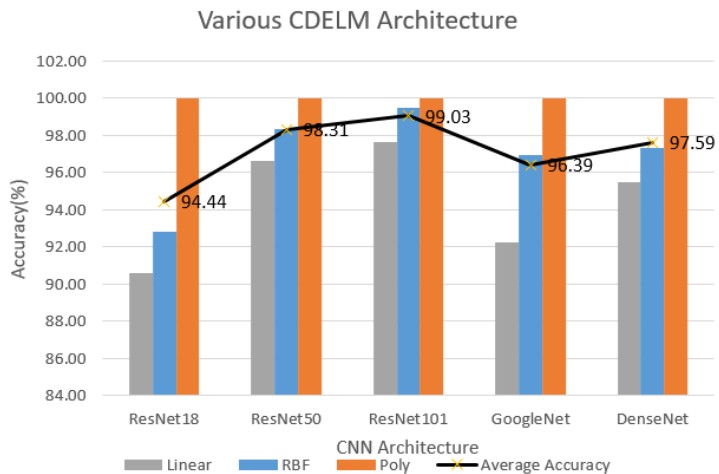

**Figure 11.** Results of Experiment on 2 class MESSIDOR Dataset.

Based on Figure 11, the highest accuracy obtained is 100% by using a polynomial kernel on each CNN architecture. It shows that using a polynomial kernel on the 2-class MESSIDOR data can separate the two classes very well. ResNet-101 architecture can represent the characteristics of the normal fundus and DR images with the best accuracy value of 99.03%, followed by ResNet-50, DenseNet, GoogleNet, and ResNet-18 architectures with an average accuracy of 98.31%, 97.59%, 96.39%, and 94.44%. The overall results of the experiment show that the more features generated from the feature extraction process using several CNN architectures, the higher the accuracy value obtained. The results of the experiment conducted on 4-class DRIVE and MESSIDOR dataset are shown in Tables 4 and 5.

The classification of 4 class DRIVE data shows poor results on the ResNet-18, ResNet-50, and GoogleNet architectures with linear kernel functions. Compared to the depth of layer complexity, the three architectures are no more profound than the ResNet-101 and DenseNet architectures. It shows that the features produced by the three architectures do not represent the features of the fundus image and cannot be divided linearly. In contrast, the RBF kernel function and polynomial in each architecture produce an accuracy above 90%, which indicates the compatibility of the RBF kernel and polynomial functions to the fundus image features. The duration of the training time in the experiment using the 4- class DRIVE dataset is relatively short, ranging from 2 to 2.5 min. The graph on the accuracy value comparison from 4-class DRIVE dataset is shown in Figure 12.

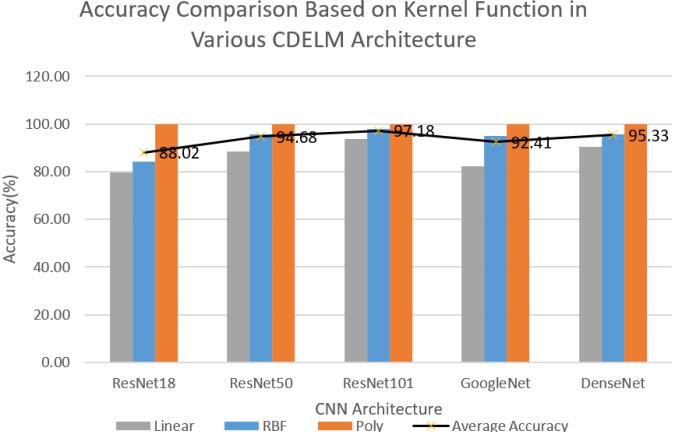

**Figure 12.** Results of Experiment on the 4 class DRIVE Dataset.

Based on Figure 12, the entire experiments using the polynomial kernel obtain perfect results, which reach 100% accuracy. The DELM classification system using a linear kernel only produces an accuracy of 79% to 93%. Meanwhile, the experiment using the RBF kernel produces higher results than that using the linear kernel, with an accuracy of 84% to 95%. The best CNN architecture in extracting fundus image features in terms of the average accuracy value is the ResNet-101 architecture, with an average accuracy of 97.18%. The next best average accuracy result is 95.33% on DenseNet architecture, followed by ResNet-50, GoogleNet, and ResNet-18 architectures with an average accuracy of 95.33%, 94.68%, 82.42%, and 88.02% consecutively.

Based on Table 5, the performance of the CDELM method on the 4-class MESSIDOR data is not as good as the CDELM performance on the 2-class MESSIDOR dataset and the 4-class DRIVE data. MESSIDOR data cannot be appropriately classified using the CDELM method with an accuracy of less than 70% on the linear kernel and RBF. The accuracy results in the experiment show that the fundus image features in the 4-class MESSIDOR dataset do not have significant differences between classes, so the DELM method cannot properly separate the data for each class. However, implementing the CDELM method using a polynomial kernel can obtain a pretty good classification result, which is more than 90% in each experiment, and the best result is 98.20%. The classification system using a polynomial kernel has a good accuracy value in every CNN architecture experiment with a relatively short training duration. The graph on the accuracy value comparison from the 4-class MESSIDOR dataset is shown in Figure 13.

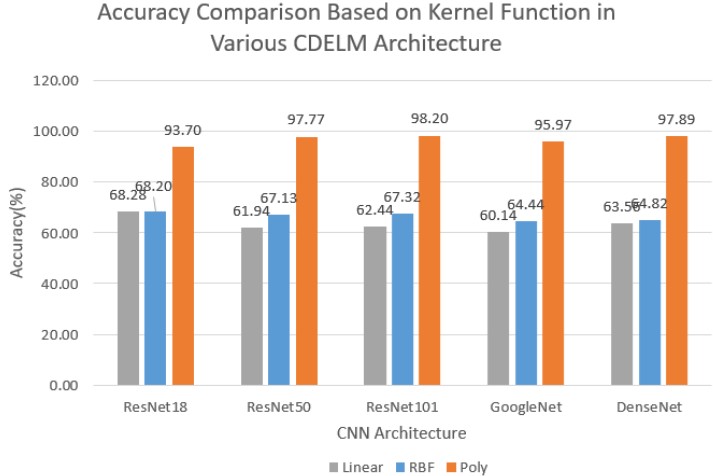

**Figure 13.** Results of Experiment on 4 class MESSIDOR Dataset.

Figure 11 shows that in each CNN architecture test, the polynomial kernel has a good performance in the 4-class MESSIDOR data classification. The accuracy of the classification system using the polynomial kernel function significantly differs significantly from that using the linear kernel and the RBF kernel. The difference in accuracy values in the linear kernel reaches 25% to 35%, while in the RBF kernel, it reaches 25% to 33%. The results of the performance comparison of each CNN architecture show that ResNet-101 is superior to other CNN architectures with an accuracy of 98.20%, followed by DenseNet, ResNet-50, GoogleNet, and ResNet-18. CNN architectures (GoogleNet, ResNet18, ResNet50, ResNet101, and DenseNet) have different extraction computation concepts. It certainly affects the value of feature extraction, which is used as data input in the classification process. Based on the overall results above, the selection of a good image extraction method greatly affects the evaluation results of the classification system. The results of the comparison of each CNN architecture can be seen in Figure 14.

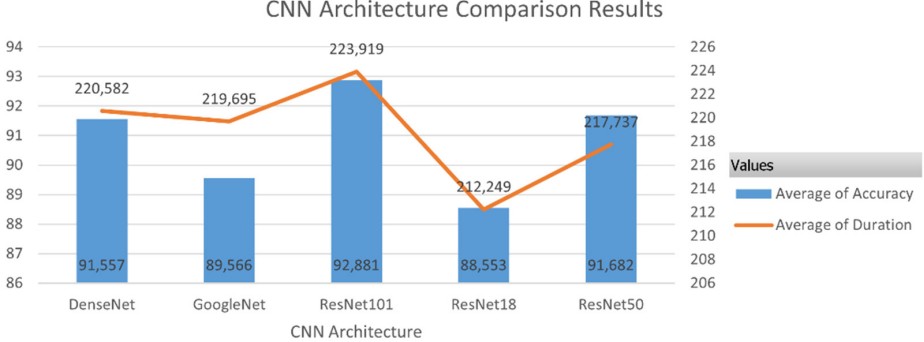

**Figure 14.** CNN Architecture Comparison Results.

Several CNN architectures were tested to well determine the results of the DR classification. Based on Figure 14, it shows that the highest overall values are obtained by the ResNet101 architecture. The average accuracy is 92.88%, the sensitivity is 92.88%, and the specificity is 92.84%. However, in computing time, ResNet101 takes the longest time compared to other architectures, with an average time of 223.92 s. The difference in accuracy values for each architecture is only approximately 4%, and the time difference is only approximately 10 s. It shows that the CNN architecture has almost the same performance. However, in the experiment conducted on 4 class data, it shows significant differences on CNN architectures. ResNet101 is a CNN architecture that has the best performance in classifying DR. The comparison of the average accuracy based on the kernel function in the whole experiment is shown in Figure 15.

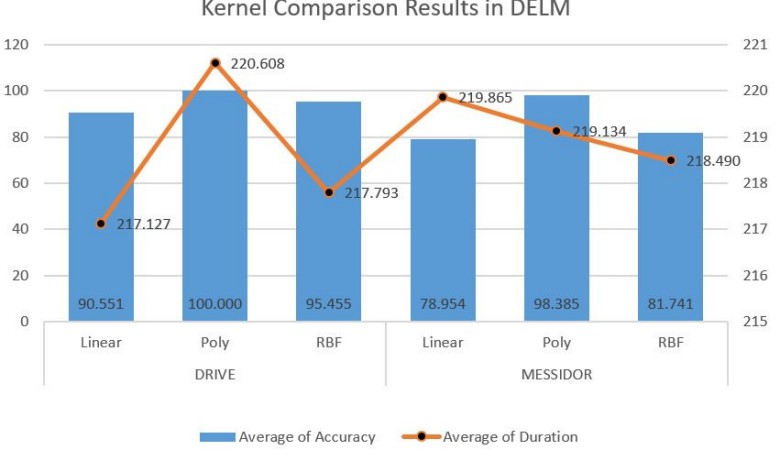

**Figure 15.** Kernel Comparison Results in DELM.

This study experimented with various kernels on the DELM model, namely linear, polynomial, and RBF. The results of Figure 15 show that the polynomial kernel type achieved the best results in every type of class (2 class and 4 class). In the 4-class experiment, the average evaluation value for each kernel decreased by 11.6% for linear, 1.6% for polynomial, and 13.8% for RBF. The smallest reduction number was the kernel polynomial. From all DELM experiments, polynomials can separate the features generated from each architecture very well. It is because the features obtained from the CNN feature extraction process have complex features. In contrast, the polynomial is more suitable for classifying global features than the RBF method. The linear kernel produces a lower accuracy than the RBF kernel, and the polynomial indicates that the fundus feature data cannot be separated linearly. Meanwhile, the three kernels have a time difference of 1 to 5 s in computing time. Therefore, the selection of the DELM kernel type has less effect on the computation time.

The number of data highly affects CDELM performance. This study conducted several experiments by taking the number of datasets on the Messidor dataset. The first experiment was conducted on the number of data according to the minimum class in the DRIVE dataset. The results of the first experiment were obtained from the best results in Table 6, that is classification using ResNet101 feature extraction with polynomial kernel on DELM. The second experiment used the same architecture as the first one, but the there were 20,000 data. The third experiment used a total of 50,000 data. Furthermore in the fourth experiment, the number of data taken was 60,000. The experiment is limited to 60,000 data collection because DELM has limitations on large number of data. When the data is very large or more than 60,000 data, multiplying a large number of square matrices in DELM causes errors during the training process.

**Table 6.** CDELM performance based on number of data.

| Experiment | Number of Training Data | Number of Testing Data | Accuracy (%) | Sensitivity (%) | Specificity (%) | Duration (s) |
|---|---|---|---|---|---|---|
| Ex 1 | 16,000 | 4000 | 98.27 | 98.27 | 98.28 | 165.84 |
| Ex 2 | 32,000 | 8000 | 98.75 | 98.75 | 98.75 | 233.91 |
| Ex 3 | 40,000 | 10,000 | 99.49 | 99.49 | 99.49 | 2215.39 |
| Ex 4 | 48,000 | 12,000 | 99.58 | 99.58 | 99.58 | 5555.17 |

The results of the four experiments in Figure 16 show that the more data, the better the classification system. The value of accuracy increases in line with the higher computational time required in the classification process. From the comprehensive test, ResNet-101 has a good performance. Compared to conventional ResNet-101 with 64 batch size, ResNet101-DELM has a better performance and a shorter training time duration. It is evidenced by the experiment to compare the performance of ResNet101 and ResNet101-DELM, as shown in Table 7.

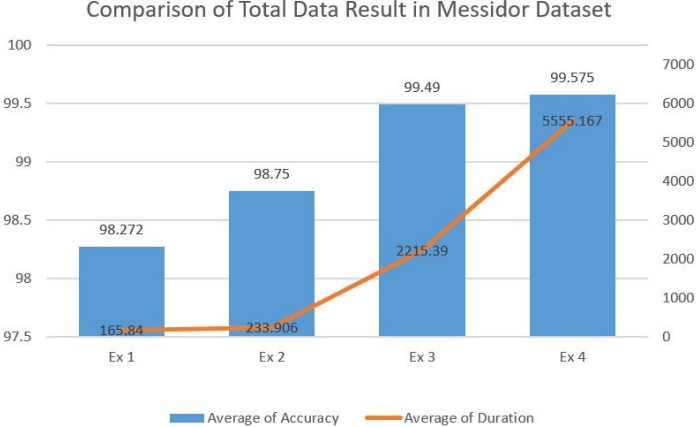

**Figure 16.** Results of Total Data Comparison on MESSIDOR Dataset.

**Table 7.** Performance comparison of ResNet101 and ResNet101-DELM.

| | | **ResNet-101** | | | |
|---|---|---|---|---|---|
| | Dataset | Accuracy (%) | Sensitivity (%) | Specificity (%) | Duration (s) |
| | DRIVE | 99.93 | 99.93 | 99.93 | 599.00 |
| 2 Class | MESIDOR | 92.99 | 92.99 | 93.05 | 594 |
| | | ResNet-101-DELM | | | |
| | DRIVE | 100.00 | 100.00 | 100.00 | 303.15 |
| | MESIDOR | 100.00 | 100.00 | 100.00 | 293.08 |
| | | **ResNet-101** | | | |
| | Dataset | Accuracy (%) | Sensitivity (%) | Specificity (%) | Duration (s) |
| | DRIVE | 100.00 | 100.00 | 100.00 | 1149.00 |
| 4 Class | MESIDOR | 91.40 | 91.40 | 91.49 | 1123.00 |
| | | ResNet-101-DELM | | | |
| | DRIVE | 100.00 | 100.00 | 100.00 | 142.72 |
| | MESIDOR | 98.20 | 98.20 | 98.19 | 166.94 |

Based on the results of a performance comparison of ResNet101 and ResNet101-DELM in Table 7, the graphs on comparing the accuracy and duration of training time for 2 class and 4 class are shown in Figure 17.

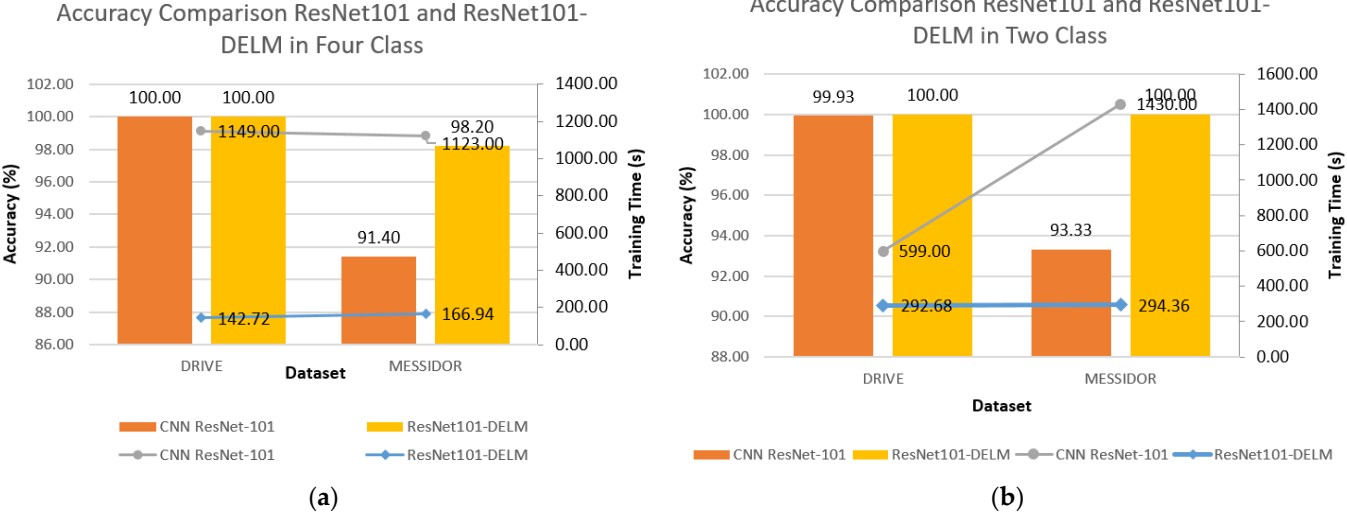

(**a**)            (**b**)

**Figure 17.** (**a**) Accuracy comparison of ResNet101 and ResNet101-DELM on 4 Class (**b**) Accuracy comparison of ResNet101 and ResNet101-DELM on 2 Class.

Based on all experiments, ResNet101-DELM produces a higher accuracy than conventional ResNet-101 in which the accuracy values reach almost 100% in several experiments. Combining CDELM with ResNet101 can increase accuracy by 0.01% on the DRIVE data and up to 7% on the MESSIDOR data. In addition, the ResNet101-DELM method requires a much shorter time than conventional ResNet-101, with a time difference of 300 to 1000 s. It shows that ResNet101-DELM is more optimal than conventional ResNet-101. A comparison of the results of the evaluation of the fundus image classification system in identifying diabetic retinopathy from several previous studies is shown in Table 8.

**Table 8.** Performance comparison of ResNet101 and ResNet101-DELM.

| Method | Dataset | Accuracy (%) | Sensitivity (%) | Specificity (%) | Duration (s) |
|---|---|---|---|---|---|
| 5-Layered CNN [46] | MESSIDOR | 98.15 | 98.94 | 97.87 | - |
| Modified Alexnet [12] | MESSIDOR | 92.35 | - | 97.45 | - |
| ResNet-101 [47] | DRIVE | 95.10 | 79.30 | 97.40 | - |
| CLAHE+ ResNet-101-DELM | DRIVE | 100.00 | 100.00 | 100.00 | 142.72 |
| CLAHE+ ResNet-101-DELM | MESIDOR | 98.20 | 98.20 | 98.19 | 166.94 |

Based on Table 8, the ResNet101-DELM hybrid performed well in fundus image classification for diabetic retinopathy identification. Research [46], by customizing five layers of CNN and segmentation process resulted in an accuracy value of 98.15% on the MESSIDOR dataset. Research [12] modifies the CNN Alexnet architecture with input images using only green channels; on MESSIDOR data this study produces an accuracy of 92.35%. Research [47] using the ResNet101 method on the DRIVE dataset only obtained an accuracy of 95.10%, while in this study using the same method and the addition of CLAHE an accuracy of 100% was obtained. With the same accuracy, ResNet101-DELM has a much shorter training time. The sensitivity value shows how the classification system identifies normal fundus as DR. While specificity identifies the fundus image in a DR Class (Mild, Moderate, Severe) as a normal class; in medical cases, a classification system with a higher sensitivity value is more efficient than a specificity value because when a normal fundus is identified as DR, it will increase the patient's awareness about DR. Based on a comparison with several previous studies, CNN-DELM is an image classification method that has good performance and ashort training time. The good performance of CNN-DELM in image classification has a weakness, which is in the classification of large amounts of data, so data partitioning is needed to calculate the dimensions of the square matrix in the DELM method.

## 4. Conclusions

In this research, classifications were made with two multi-class experiments: 2 class (Normal and DR) and 4 class (Normal, Mild, Moderate, and Severe). The initial stage of the research was cropping the image, CLAHE; resizing according to the CNN architecture input size; and performing the augmentation process. This study experimented with various kernels on the DELM model, namely linear, polynomial, and RBF. The results of Figure 13 show that the polynomial kernel type achieved the best results in every type of class (2 class and 4 class). In the 4-class experiment, the average evaluation value for each kernel decreased by 11.6% for linear, 1.6% for polynomial, and 13.8% for RBF. The smallest reduction number was the kernel polynomial. From all DELM experiments, polynomials can separate the features generated from each architecture very well. It is because the features obtained from the CNN feature extraction process have complex features. In contrast, the polynomial is more suitable for classifying global features than the RBF method. The linear kernel produces lower accuracy than the RBF kernel, and the polynomial indicates that the fundus feature data cannot be separated linearly. Meanwhile, the three kernels have a time difference of 1 to 5 s in computing time. Based on all experiments, ResNet101-DELM has better accuracy than other CNN architecture. Compared with conventional ResNet-101, ResNet101-DELM produces a higher accuracy in which the accuracy values reach almost 100% and a faster training time. So, the hybrid method DELM is more optimal than conventional CNN. The limitation of the CDELM method is a large number of data. When the data is very large or the data is more than 60,000, multiplying a large number of square matrices in DELM causes errors during the training process.

**Author Contributions:** Conceptualization was carried out by D.C.R.N., F.F. and R.H.; writing was carried out by D.C.R.N.; data curation was carried out by D.C.R.N.; supervision was carried out by Fatmawati, and R.H. H.R. was the reviewer. M.I.H. was the curator. R.N. carried out visualization. Original draft preparation was carried out by A.A. and R.A.S. A.P. carried out the formal analysis. All authors have read and agreed to the published version of the manuscript.

**Funding:** This research received no external funding.

**Data Availability Statement:** This research uses two types of diabetic retinopathy datasets: the DRIVE (https://drive.grand-challenge.org/, accessed on 27 November 2022) and MESSIDOR (https://www.adcis.net/en/third-party/messidor/, accessed on 27 November 2022) datasets.

**Conflicts of Interest:** The authors declare no conflict of interest.

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
