# Peer review of "Image Fundus Classification System for Diabetic Retinopathy Stage Detection Using Hybrid CNN-DELM"

_2504-2289, doi:10.3390/bdcc6040146_

Round 1

Reviewer 1 Report

Article is about Diabetic Retinopathy(DR) detection using hybrid CNN- DELM method.

Author have not highlighted the specific research challenges in DR detection and RQs for this study.

Literature review on DR detection using CNN methods sounds like summary of existing work, there is no critical analysis on methodology and results. Authors need to include the research gaps in existing methods and explain how the proposed methods can overcome the gaps.

In materials and methods section, the methodology can be further improved by illustrating the different phases of DR detection based on fundus images in generic manner. Author can move the figure 2 while explaining the different phases on DR detection in the study. Sample fundus images can be illustrated in the section 2.1.

Section 2.2 , the explanation on CNN architecture is limited, not clear, must explain the feature learning methods: pooling and CNN+ ReLu.

Architectures in the CNN algorithm for this study includes include Google Net, Res- 15 Net, and DenseNet, how different is each architecture must be explained with illustrations.

DELM architecture explanation is sufficient.

Results are promising for all architectures and DRIVE and MESIDOR datasets. Author must include the ratio of training and testing datasets for this study. There is no sample fundus image shown in this article.

Author must add discussion section to explain in detail the comparative result analysis with existing DR detection literature works on kernel, sensitivity, specificity, accuracy and duration.

Must check the grammar.

Author Response

  1. the CNN algorithm performed well in the classification process and had many layers. A large number of layers in the CNN algorithm required a computer with a large capacity and a long duration of the training process [16]. Several other researchers have tried to develop CNN to overcome these problems by changing the existing classification system on CNN to form a developed method that uses the convolutional features in the CNN architecture but uses a different classification method. The development of CNN methods, such as the Convolutional Extreme Learning Machine (CELM), which uses the convolutional features in the CNN architecture, and the Extreme Learning Machine classification method. A research on CELM was conducted to identify handwritten using MNIST data [17–19]. The results of this study indicated that the accuracy obtained by CELM was better than ELM and CNN, that is 98.43%, with a training time faster than CNN and ELM. Although CELM is better than CNN, basically, ELM is still a single hidden layer method and is still not good at pattern recognition in big data, so the development of the ELM method by applying a multilayer and deep learning system is called Deep Extreme Learning Machine (DELM) [20]. The DELM method has several advantages, especially in training time, making it one of the deep learning methods with the fastest training process. DELM also has good results in terms of image classification (MNIST database, CIFAR-10 dataset, and Google Streetview House Number dataset) with an average accuracy of 95.16% [21]. The DELM algorithm can produce high accuracy in just 9.02 seconds [22]. DELM is a combination of several algorithms which are the result of the development of the Extreme Learning Machine (ELM) algorithm. DELM has a more complex structure than ELM, but DELM can train models faster than the ELM algorithm
  2. Thank you for your suggestion, we’ve explained the different phases of DR detection based on fundus images.
  3. Thank you for your suggestion, we’ve already describe convolution layer, pooling, and ReLU layer in section 2.2
  4. Thank you for your suggestion, we’ve already add the illustration of each architecture
  5. Thank you for your suggestion, we’ve already add the description of the data in result section and table 4. The data is divided into training and testing data using 5-fold cross-validation. In the two classes dataset, training data contains 2880 images in each class and 720 images in each class for testing data. While in the four classes dataset, training data has 2607 images in each class and 651 images in each class for the testing dataset.
  6. Thank you for your suggestion, we’ve already explain kernel, sensitivity, specificity, accuracy, duration, and compare with other previous research in table 8.

Reviewer 2 Report

The authors presented Image Fundus Classification System for Diabetic Retinopathy 2 Stage Detection Using Hybrid CNN-DELM.

The approach seems valid & interesting results are interesting. However, I have the following minor corrections.

State of art not fully explored, authors need to cite the latest articles & compare results on the same benchmark dataset to show there is some room for importance yet.

Highlight issues with previous techniques.

Cite the following latest state of art current reported techniques & compare results with them.

Hassan, S. A., Akbar, S., Rehman, A., Saba, T., Kolivand, H., & Bahaj, S. A. (2021). Recent Developments in Detection of Central Serous Retinopathy through Imaging and Artificial Intelligence Techniques–A Review. IEEE Access.

Saba, T., Akbar, S., Kolivand, H., & Ali Bahaj, S. (2021). Automatic detection of papilledema through fundus retinal images using deep learning. Microscopy Research and Technique, 84(12), 3066-3077.

Ahmad, M. F., Akbar, S., Hassan, S. A. E., Rehman, A., & Ayesha, N. (2021, November). Deep Learning Approach to Diagnose Alzheimer’s Disease through Magnetic Resonance Images. In 2021 International Conference on Innovative Computing (ICIC) (pp. 1-6). IEEE.

Additionally, present a pseudo-code of their method.

In the discussion section, please include a confusion matrix.

Author Response

Thank you for your suggestion, we've already add the reference as your suggestion that related to our research

Reviewer 3 Report

The present paper describes a well documented research. The Introduction it well constructed on a solid base documentation of the present state of the art. The authors contribution it is described in a understanding matter which provides explanation of all the stages that were fulfilled in order to obtain relevant results within specific domain of interest. I recommend that, this paper to be published as it is.

Author Response

Thank you so much for your review.

Reviewer 4 Report

In this manuscript, the authors present a hybrid CNN-DELM for detection of the severity of diabetic retinopathy. Several CNN architectures were used for comparison of performance of classification, further enhanced by the DELM algorithm. The authors also study the effect of using kernel functions for feature extraction. Experimental results demonstrate the superiority of the CNN-DELM hybrid method over the traditional CNN methods. 

In general, the manuscript is well-written and well-commented. The results seem valid. I have the following comments/suggestions:

1) 1 It would be better to add graphical plots of training, validation, and loss curves.

2) It would be better to add a table of number of training, validation, and test datasets, and training time for the different networks for better readability. Right now the information of the actual number of datasets used seem to be missing.

3) The number of training, validation, and test data and batch size would be better if put to a power of two.

4) What kind of activation function was used for the CNN?

5) Need to justify why only augmentation using rotations were used? Why not other kind of deformations?

Author Response

  1. Thank you for your suggestion, It cannot show the training progress, cause DELM is feedforward, not an iterative methods, and matrix base concept, so it just show one value of error at the end.
  2. Thank you for your suggestion, we’ve already add the description about number of data training and testing in result section and table 4.
  3. Thank you for your suggestion, in this research batchsize only used in conventional ResNet-101 and its already explained in result section.
  4. Thank you for your suggestion, in this research we use ReLU as an activation function in CNN, its explained in section 2.2
  5. Thank you for your suggestion, The rotation method is suitable if applied to images with features like circles. The fundus image has circular features, so applying the rotation method for the augmentation process is more efficient (its explained in result section)

Round 2

Reviewer 1 Report

Revised version is statisfactory